# Encapsulation of Adipose-Derived Mesenchymal Stem Cells in Calcium Alginate Maintains Clonogenicity and Enhances their Secretory Profile

**DOI:** 10.3390/ijms21176316

**Published:** 2020-08-31

**Authors:** Lucille Capin, Nacira Abbassi, Maëlle Lachat, Marie Calteau, Cynthia Barratier, Ali Mojallal, Sandrine Bourgeois, Céline Auxenfans

**Affiliations:** 1Banque de Tissus et de Cellules des Hospices Civils de Lyon, Hôpital Edouard Herriot, Place d’Arsonval, 69003 Lyon, France; abbassi.nacira@gmail.com (N.A.); maelle.lachat@gmail.com (M.L.); calteaumarie@yahoo.fr (M.C.); 2Univ Lyon, Université Claude Bernard Lyon 1, LAGEPP UMR 5007 CNRS, F-69100 Villeurbanne, France; cynthia.barratier@univ-lyon1.fr (C.B.); sandrine.bourgeois@univ-lyon1.fr (S.B.); 3Univ Lyon, Université Claude Bernard Lyon 1, ISPB-Faculté de Pharmacie, F-69008 Lyon, France; 4Service de chirurgie plastique, reconstructrice et esthétique, Hôpital de la Croix Rousse, Hospices Civils de Lyon, 69004 Lyon, France; alain-ali.mojallal@chu-lyon.fr; 5Univ Lyon, Université Claude Bernard-Lyon 1, 8 avenue Rockefeller, 69008 Lyon, France

**Keywords:** adipose-derived mesenchymal stem cells, encapsulation, alginate, secretome, cell therapy, microparticles

## Abstract

Adipose-derived mesenchymal stem cells (ASCs) are well known for their secretory potential, which confers them useful properties in cell therapy. Nevertheless, this therapeutic potential is reduced after transplantation due to their short survival in the human body and their migration property. This study proposes a method to protect cells during and after injection by encapsulation in microparticles of calcium alginate. Besides, the consequences of encapsulation on ASC proliferation, pluripotential, and secretome were studied. Spherical particles with a mean diameter of 500 µm could be obtained in a reproducible manner with a viability of 70% after 16 days in vitro. Moreover, encapsulation did not alter the proliferative properties of ASCs upon return to culture nor their differentiation potential in adipocytes, chondrocytes, and osteocytes. Concerning their secretome, encapsulated ASCs consistently produced greater amounts of interleukin-6 (IL-6), interleukin-8 (IL-8), and vascular endothelial growth factor (VEGF) compared to monolayer cultures. Encapsulation therefore appears to enrich the secretome with transforming growth factor β1 (TGF-β1) and macrophage inflammatory protein-1β (MIP-1β) not detectable in monolayer cultures. Alginate microparticles seem sufficiently porous to allow diffusion of the cytokines of interest. With all these cytokines playing an important role in wound healing, it appears relevant to investigate the impact of using encapsulated ASCs on the wound healing process.

## 1. Introduction

Several diseases, such as osteo-articular disorders, diabetes, cancers, cardiovascular diseases, and skin disorders, could be treated using adipose-derived mesenchymal stem cells (ASCs), and these cells have been used in many studies [1,2,3,4,5,6]. They represent an alternative to bone marrow-derived stem cells in cell therapy applications. Indeed, ASCs have several advantages over bone marrow-derived cells: They are easier to extract in greater volumes and have a higher proliferative potential [7,8,9,10,11,12]. ASCs also have a significant secretory potential, producing factors that play a crucial role in tissue repair [13,14,15,16], in particular angiogenic, anti-inflammatory, antioxidant, antifibrotic, and/or antiapoptotic cytokines [17,18,19,20]. However, stem cells expanded in culture before in vivo transplantation have a short survival duration, thus reducing their therapeutic potential [21,22]. To potentialize the therapeutic effect of injected ASCs, various strategies have been tested (hypoxic culture conditions, genetic modification, 3-D culture, stromal stem cell aggregates, scaffold) [23,24,25]. The strategy with the most potential appeared to us to be microencapsulation of ASCs in biomaterials as it is compatible with long-duration cellular survival [26,27]. Indeed, this improved survival would allow the use of ASCs in humans for a wound healing purpose [28].

Cellular microencapsulation consists in surrounding cells with a biocompatible polymer layer to form microparticles. The polymers tested so far have been semi-permeable hydrogels, allowing the diffusion of low-molecular-weight molecules, such as nutrients and oxygen, from the external medium to the enclosed cells, and of small proteins with therapeutic potential produced by the cells to the surrounding medium. This system has the advantage of protecting the cells from their environment, in particular in the case of allogeneic cells, by preventing antibodies and immune cells from coming into contact with them and causing their destruction [21,29]. It thus improves the safety of cellular therapies [30]. Several techniques are currently available for microencapsulation of cells, differing based on the size of the particles, whether they form spherical particles, their viscosity, and the desired production rate. For example, encapsulation techniques include coaxial flow [31,32], electrostatic potential [33,34], vibration [35], jet cutting [36], microfluidics [37], and coacervation [38]. For this study, we formed microparticles by prilling vibration, which produces spherical microparticles when the cell-polymer suspension passes through a nozzle. The vibrating system atomizes the cellular suspension as droplets, which, upon contact with a jellification solution, form microparticles. The prilling vibration encapsulation equipment used here (Encapsulator B-390 ^®^, Buchi) is compatible with medical-grade production, in line with good manufacturing practices (GMP). This is a considerable advantage and should be considered for clinical applications. In addition, all the encapsulation steps can be performed in sterile conditions.

Biocompatible polymers with potential for use in encapsulation include pectin [39], agarose [40], chitosan [41,42], and alginate. We chose to use a medical-grade alginate here as it is cheap, can be sterilized, and has been the most widely studied in cellular encapsulation applications [27,29,43,44,45,46].

For this study, cultured ASCs were encapsulated. We first assessed the morphometric stability of the ASC-containing calcium alginate microparticles for up to 16 days post-encapsulation, then we tested their capacity to maintain cellular viability and metabolic activity (Trypan blue, MTT assay (3-(4,5-dimethylthiazol-2-yl)-2,5-diphenyltetrazolium bromide)), their proliferative potential (population doubling and doubling time), their stem cell characteristics by the clonogenicity test (colony forming efficiency), their capacity to differentiate into the three standard lineages, and their phenotypes (CD73, CD90, CD45, and CD34 expression). Finally, we used culture supernatants to study and compare the secretomes of encapsulated and non-encapsulated ASCs to determine the advantages of encapsulation over monolayer culture.

## 2. Results

### 2.1. Morphometric Analysis of Microparticles

The morphology and diameter of microparticles were compared on D0 and D16 after encapsulation for three donors at P3, and one donor at P3 and P4 (Figure 1). On D0, the microparticles obtained were round, well-delimited, not drop shaped, and free from cracks, Figure 1a. After 16 days of culture, the particles presented a few signs of fragility, such as cracks, had less clearly defined borders, and a less rounded morphology, as shown in Figure 1b. Nevertheless, during the 16 days of culture, no cell was observed growing outside the microparticles, but after 16 days, the apparent fragility correlated with cellular escape, and few cells were found in culture flasks beyond this period.

The diameter of the microparticles obtained with the 150-µm nozzle was measured using Image J software on 20 microparticles from each donor at D0 and 20 microparticles from each donor at D16, as shown in Figure 1c. These mean diameters were compared using the paired Wilcoxon test and were not statistically different (*p* = 0.3125). Mean diameters of 538.44 ± 72.22 and 519.38 ± 84.79 µm were obtained on D0 and D16, respectively.

Alginate polymers are therefore stable, retaining cells for at least 16 days in microparticles of a constant size.

### 2.2. Viability and Metabolic Activity of Encapsulated ASC

The viability of encapsulated cells from two donors at P3 and one donor at P3 and P4 was assessed over 16 days of culture, as shown in Figure 2a. On D0 after encapsulation, the mean viability was 77% ± 3%. After 16 days of culture, the mean viability was 74% ± 4%. The cells therefore remain viable inside microparticles for at least 16 days after encapsulation. The mean viability from each day remains not statistically different compared to D0 (Mann and Whitney test). Thus, alginate particles supported the diffusion of nutrients, vitamins, and glucose essential for survival of the encapsulated ASCs. In comparison, the mean viability of monolayer cells was 97% ± 3% on D0 and 98%± 3% on D16.

The metabolic activity of cells obtained after dissolving alginate microparticles was compared to that of monolayer cells using an MTT test, as shown in Figure 2b, on cells from three donors over 7 days. Interestingly, the metabolic activity measured for monolayer cells increased from D0 to D5 and D0 to D7, indicating that cells were proliferating during this period. In contrast, the comparison of metabolic activity of encapsulated cells by the Mann and Whitney test remains not statistically different between D0 and D2 (*p* = 0.171), D5 (*p* = 0.171) and D7 (*p* = 0.443), and between D2 and D5 (*p* = 0.343), D2 and D7 (*p* = 0.243), and D5 and D7 (*p* = 0.100). These results suggest that encapsulated cells remain metabolically active for at least 7 days, but their proliferation is limited, probably due to the size of the microparticle. Besides, the number of cells seeded at D0 for both encapsulated and monolayer cells was the same (0.3 million cells/well) and yet the metabolic activity of encapsulated cells was higher than monolayer cells, suggesting that encapsulation improves metabolic activity at D0. Otherwise, no statistical differences were observed at D2 (*p* = 0.100), D5 (*p* = 0.0571) and D7 (*p* = 0.343) between the two conditions.

### 2.3. Clonogenic and Proliferative Potential of ASCs after Encapsulation

The clonogenic potential, or colony forming efficiency (CFE), and the proliferative potential of ASCs from two donors at P3 and one donor at P3 and P4 were studied after 7 days’ encapsulation and compared to those measured for cells grown as monolayers (Figure 3).

To be useful in clinical applications, ASCs must have a CFE greater than or equal to 1% upon extraction and greater than or equal to 5% after culture. However, the CFE is very variable between donors. CFE measured for cells grown as a monolayer and for encapsulated cells, after dissolving the capsule on D7 post-encapsulation, were not statistically different, as shown in Figure 3a. For the three donors, the mean CFE obtained for monolayer cultures and encapsulated cells was respectively 13.1% ± 3.22% and 10.0% ± 5.96%. CFE obtained from monolayer cells were not significatively different than CFE from encapsulated cells (*p* = 0.245) according to the Mann and Whitney test.

The proliferative potential was determined by measuring the population doubling (PD) and determining the doubling time (DT) for cells from two donors at P3 and one donor at P3 and P4, as shown in Figure 3b,c. Sub-confluency was obtained after 10 days on average for both monolayer and encapsulated cells. The monolayer cell population doubled 2.94 times ± 0.507 during the culture period against 3.29 times ± 0.784 for the encapsulated cell population. Besides, the doubling time obtained for monolayer cells was 0.283 ± 0.097 division per cells during the culture period against 0.298 ± 0.085 division per cells for encapsulated cells. Thus, the results showed no significant difference in PD (*p* = 0.149) and DT (*p* = 0.333) when compared using the Mann and Whitney test. Encapsulation in the conditions used here did not alter the proliferative properties of ASCs upon return to culture.

### 2.4. Pluripotential

To determine whether encapsulation interferes with the capacity of cells to differentiate into the three recommended lineages [47,48], ASCs from two donors were exposed to media inducing differentiation into chondrocytes, osteocytes, and adipocytes 1 day after encapsulation (Figure 4). Monolayer-cultured ASCs from the same two donors were used as control cells. For each donor, chondrocyte, Figure 4a, osteocyte, Figure 4b, and adipocyte, Figure 4c,d, induction was tested in two conditions: On encapsulated cells (condition 1) and on cells that were initially encapsulated and were returned to monolayer culture after dissolving the microparticles (condition 2). All formerly encapsulated ASCs (condition 2) from the three donors differentiated into the three mesenchyme lineages, as revealed by the Alcian Blue (chondrocytes), Alizarin Red (osteocytes), and Oil red O (adipocytes) staining patterns. For condition 1, where cells were still encapsulated, differentiation was observable only for the adipocyte lineage. Indeed, Alizarin Red staining to detect calcium deposits also stained the calcium alginate capsule, making the results impossible to interpret. For chondrocyte differentiation, the differentiation process requires the cells to be pelleted, which is impossible with microparticles.

The experiment performed in condition 2 shows that encapsulated cells retain their capacity to differentiate into adipocytes, osteocytes, and chondrocytes.

### 2.5. Characterization of Adipose-Derived Stem Cells after Encapsulation

The cell surface profile was assessed on encapsulated cells after dissolving alginate microparticle and on monolayer cells from two donors (16,023 at P3 and 17,149 at P3). After encapsulation for 1 day, flow cytometry (Figure 5) revealed that ASCs from the two donors express the classical mesenchyme markers CD73 and CD90 [49], and lack the hematopoietic marker CD45, and CD34, which is a native ASC marker that gradually disappears as the passage number increases [50]. Cell surface profiles of encapsulated and monolayer ASCs were similar. These results indicate that ASC encapsulation in alginate for 1 day does not alter the phenotype of cultured ASCs.

### 2.6. Secretome of Encapsulated ASCs

Secretion of cytokines IL-6 (interleukin-6), IL-8 (interleukin-8), and VEGF (vascular endothelial growth factor) was measured by quantitative ELISA in supernatants from encapsulated ASCs and monolayer cultures for seven donors at P3 (with one donor at P3 and P4) after culture in serum-free medium for 48 h, as shown in Figure 6a.

Encapsulated ASCs consistently produced greater amounts of IL-6, IL-8, and VEGF compared to monolayer cultures. Alginate microparticles are therefore sufficiently porous to allow diffusion of the cytokines of interest.

The multiplex qualitative ELISA revealed data on the secretion of additional cytokines for four donors at P3, as shown in Figure 6b. In this assay, encapsulated cells secreted MCP-1 (monocyte chemoattractant protein 1), TGF-β1 (transforming growth factor β1), and MIP-1β (macrophage inflammatory protein-1β). MCP-1 was the only cytokine detectable for the monolayer culture of cells from donor 18,019; the other monolayer cultures failed to express detectable levels of any cytokine. ASC encapsulation therefore appears to enrich the secretome with new cytokines not synthesized by monolayer cultures. In this qualitative assay, IL-6 was detectable in the secretome from encapsulated ASCs from donor 18,011, and it was undetectable in the quantitative analysis. This apparent discrepancy could be the result of a higher sensitivity of the qualitative ELISA. Unfortunately, we could not get the same kind of data for IL-8 because this cytokine is not one of those detectable by the multiplex qualitative ELISA.

For encapsulated cells from donor 16,023 at P3, secreted levels of IL-6 and VEGF were higher than for monolayer cells (3.17-fold higher for IL-6, 1.27-fold higher for VEGF). In contrast, for this donor, encapsulated cells appeared to secrete less IL-8 than monolayer cells. For cells at P3 from donors 16,198, 16,148, 17,149, 18,011, 18,019, and 18,020, encapsulated cells secreted many more cytokines or growth factors than monolayer cultures. In order to better compare secretome from monolayer cells versus encapsulated cells, we pooled the data obtained for each donor concerning IL-6, IL-8, and VEGF secretions and expressed them in pg for 100,000 cells.

## 3. Discussion

The objectives of this study were (i) to develop an encapsulation protocol for adipose stromal stem cells extracted from adipose tissue and (ii) to assess the impact of encapsulation on ASC properties. Cellular microencapsulation was developed both to reduce premature death of cells following injection and to limit their migratory capacity [24,43]. For in vivo studies, biocompatible, spherical, solid, and homogeneous microparticles are required to guarantee cell survival without altering their stem cell properties. In this study, stem cells (ASCs) were reproducibly encapsulated, using a prilling vibration method, in microparticles of alginate, a biocompatible and biodegradable polysaccharide [51,52].

Our results demonstrated the feasibility and reproducibility of the encapsulation protocol used for this study. Indeed, cell viability was greater than 70% over 16 days for the three donors tested, and according to the MTT test, measuring the degree of metabolic activity, encapsulated cells remained viable and active but proliferated little over 7 days. The size of the microparticles produced can be modified by adapting the nozzle size in the prilling system. With a 150-µm nozzle, ASCs were encapsulated in alginate particles with a mean diameter of 538.44 ± 72.22 µm. This size is compatible with passage through a 21G needle (i.d. 0.80 mm), and thus with applications involving intradermal delivery of encapsulated ASC.

Tests of clonogenicity indicated that the encapsulated cells retained their stem cell properties. Indeed, when the microparticles were dissolved and the released ASCs grown as a monolayer, they formed colonies and differentiated into adipocytes, osteocytes, and chondrocytes upon exposure to the corresponding induction media.

In clinical applications, it is important to avoid cellular escape from the injection site. In our microencapsulated culture conditions, cellular escape was limited by the semi-solid structure of the calcium alginate microparticles due to the reticulation of the alginate by the calcium ions during the encapsulation process. To maintain microparticles’ integrity, the calcium concentration in culture medium had to be adjusted, and after 16 days in culture, very few cells were found in the flasks during medium changes. Cellular escape is therefore limited during a 16-day period but exist beyond. In order to improve microparticles’ solidity, it would be interesting to test another chelating ion, such as zinc, for example.

Furthermore, preclinical studies involving injection of ASCs into cardiac muscle in rats [53,54] and pig liver [55] reported no inflammatory reaction or fibrosis at the injection sites, and similar results were reported when ASCs were systemically injected: The cells were detectable in inflammatory tissues and appeared to play a role in reducing inflammation [56,57]. Nevertheless, in human medicine, it is very important to limit the risks of triggering an immune response. As part of this, the predicted biocompatibility of the alginate used must be verified to avoid immunization of the host. Encapsulation should protect ASCs against immune responses but only if the polymer itself is not immunogenic. For this to be the case, the polymer must be ultrapure—raw alginate presents a significant number of elements that might trigger an inflammatory response [58]—to avoid necrosis of the encapsulated cells. The alginate used in this project was selected for its quality as well as its purity. It thus meets the requirements of the European pharmacopoeia, making it compatible with clinical applications.

Concerning the clinical implications of cellular escape, most clinical trials on allogeneic ASCs report few side effects directly linked to cell product. Indeed, in a study on lateral epicondylosis, the authors observed no serious adverse events following allogeneic ASC injection during the observation period in any of the participants. [59]. The same conclusion was given in a clinical trial on bilateral osteoarthritis [60] and in a study on graft versus host disease [61]. In another clinical trial using allogeneic ASCs for complex perianal fistulas in Crohn’s disease, no adverse events related to treatment or immune reactions associated with the development of donor-specific antibodies were observed. The authors also reported no association between positivity for donor-specific antibodies and therapeutic response [62]. Nevertheless, in refractory rheumatoid arthritis, Álvaro-Gracia et al. noted the generation of ASC-specific anti-Human Leukocyte Antigen-I antibodies in 19% of the treated patients but without apparent clinical consequences [63].

Otherwise, Vassalli and Moccetti qualified allogeneic mesenchymal stem cells products (both derived from bone marrow and adipose tissue) as safe in phase-I/II trials in patients with acute myocardial infection in a review on cardiac repair [64]. Besides, in their review on the safety of adipose-derived cell therapy in clinical trials, Toyserkani et al. wrote that the only adverse events occurring were rather related to the harvesting of adipose tissue or trauma associated with injection. They conclude that there was no clear evidence of a clinical immune response even if there was antibody production. Concerning the use of such therapy in patients with previous malignancy, the authors found that no data has shown an increased risk of cancer or relapse [65].

To finish, a meta-analysis on clinical trials using mesenchymal stem cells (both adipose and bone marrow derived) was unable to detect associations between stem cell treatment and the development of acute infusional toxicity, organ system complications, infection, death, or malignancy [66]. Considering all these data, the risk of alloimmunization following cellular escape seems to be relatively low.

According to the literature, direct injection of ASCs presents two main limitations. The first is excess cellular mortality due to the mechanical forces exerted during injection [67,68], and the second is the risk of cells moving out of the injection site as a result of their migratory properties [69,70]. Encapsulation overcomes these limitations and presents numerous advantages over the direct injection route. Indeed, according to our results, in addition to maintaining good cell survival, encapsulation recreates physiological conditions by mimicking the native ASC microenvironment.

In the case of administration to a human, selected ASCs would come from a single donor in order to limit possible alloimmunization. Encapsulated ASCs would be injected intradermally, and dose and dose regimens will have to be assessed. Concerning the effect of ASCs on the wound healing process, ASCs cultivated in monolayer were previously injected intradermally in a model of wound in nude mice with promising results on wound healing kinetic [28]. Therefore, we could expect that encapsulation would allow ASCs to stay longer in the derm and may enhance our previous results according to the effect of encapsulation on ASC secretome.

To further assess the potential applications of our microencapsulated ASCs, we compared the secretomes for ASC monolayers and encapsulated ASCs and found that encapsulation altered cellular secretions. Thus, encapsulated ASCs tend to secrete larger amounts of a wider variety of cytokines. Several cytokines of interest clearly emerged from the secretion profile for encapsulated ASCs. These included proinflammatory IL-6 and IL-8, which are expressed in vivo immediately after a skin lesion. They recruit and activate inflammatory cells and play a role in the migration and proliferation of keratinocytes, promoting wound healing [71]. VEGF was also detected in the secretome of encapsulated ASCs in large amounts and is evidence of their role in tissue vascularisation. In addition, the ASC secretome contained TGF-β1, a growth factor that attracts macrophages and stimulates them to produce other cytokines and growth factors contributing to the vascular phase of wound healing [71]. The cytokine MCP-1 (CCL2) was also detected; it stimulates G-protein-coupled receptors to promote chemotaxis of monocytes, lymphocytes, and neutrophils [72]. This phenomenon is part of inflammation and may be useful for wound healing. Finally, the cytokine MIP-1 β, or CCL4, is also a chemotactic agent for immune cells, in particular monocytes, macrophages, and granulocytes [73]. This secretory potential of encapsulated ASCs could be further enhanced following injection, for example, in inflammatory situations, such as following Tumor Necrosis Factor-α stimulation.

The comparison of the secretomes for different donors by quantitative and qualitative ELISA revealed heterogeneous secretory profiles, not only between donors but also between the “encapsulated” and “non-encapsulated” conditions for the same donor. Given the limited number of donors studied and the fact that patients’ medical history is not always known, it was impossible to determine the source of this heterogeneity.

Taken together, the results presented here indicate that encapsulation protects ASCs and enhances their secretory profile without altering their clonogenic properties. It therefore appears relevant now to study the impact of using encapsulated ASCs on the wound healing process.

## 4. Materials and Methods

### 4.1. Origin, Isolation, and Culture of ASCs

Written informed consent was obtained from patients (seven women aged 45.9 years old (range: 35–56), with a BMI of 24.2 kg/m² (range: 20.8–29.2)) undergoing liposuction of abdominal subcutaneous adipose tissue under general anesthesia in the department of plastic and reconstructive surgery at Lyon university hospital. Liposuction was performed with a PAL^®^ LipoSculptor™ (MicroAire Aesthetics, Charlottesville, VA, USA) using a 3-mm Coleman cannula. The volume of adipose tissue obtained was greater than 500 mL. Surgical residue was collected in line with French regulations, and our activity was declared to French ministry for research (DC n°2008-162).

To isolate the stromal-vascular fraction (SVF), after centrifugation (1962 g for 3 min), the oil (upper phase) and tumescent phases (lower phase) were removed. Adipose tissue was then digested with collagenase (0.1 U/mL, NB6 collagenase GMP-grade, Serva Electrophoresis Roche, Indianapolis, IN, USA) at 37 °C for 45 min under constant shaking. Digestion was stopped by dilution with Dulbecco’s Modified Eagle’s Medium (DMEM with glutamax, Gibco Invitrogen, Carlsbad, CA, USA) containing 10% fetal calf serum (FCS, HyClone, Logan, UT, USA). Samples were centrifuged at 300 g for 5 min, floating adipocytes were discarded, and SVF cells were pelleted, rinsed with medium, and centrifuged once again.

Freshly isolated ASCs were seeded at a density of 40,000 cells/cm² (passage 0, P0) in proliferation medium—DMEM Glutamax (Gibco), 10% of FCS (HyClone), 10 ng/mL GMP-grade fibroblastic growth factor (FGF-2 GMP-grade, Miltenyi Biotec, Bergisch Gladbach, Germany), 100 U/mL penicillin (Panpharma, Fougères, France), 100 µg/mL gentamicin (Panpharma), 5 µg/mL fungizone (Panpharma)—at 37 °C, under humidified 5% CO_2_. The medium was replaced after cells had adhered for one hour, and subsequently 3 times per week. Sub-confluent cell layers (90–95%) were detached using trypsin-EDTA (Gibco, Thermo Fisher Scientific, Waltham, MA, USA), centrifuged, resuspended in complete medium at 4000 cells/cm², and amplified until the number of cells was sufficient for encapsulation.

### 4.2. Encapsulation of Cells in Alginate

Initially, 10 g of sodium alginate (Buchi, France) was dissolved in 400 mL of an aqueous 4% (*w*/*v*) glucose solution to produce 500 mL of sodium alginate solution at 2%. ASCs from each donor were dispersed in 20 mL of this 2% alginate solution, at 4 × 10^6^ cells/mL. Then, cells were encapsulated by the prilling vibration technique using the B-390 Encapsulator (Buchi, France) fitted with a 150-µm nozzle. The laminar jet of the suspension through the 150-µm nozzle was adjusted to a flowrate of 3.1 mL/min. The vibration frequency, applied to split the laminar jet into fine droplets, was set to 366 Hz. Finally, the voltage of the electrode forming an “umbrella” at the nozzle outlet (to avoid microparticles sticking together) was adjusted to 2000 V. The droplets of sodium alginate containing encapsulated cells then encountered the aqueous jellification solution, containing 76 mM calcium chloride (Calbiochem, France), 85 mM glucose (Macopharma, France), and 6 mM HEPES (Sigma Aldrich, St. Louis, MO, USA) at physiological pH and osmolarity, where the alginate microparticles were formed.

Microparticles were rinsed with lactated Ringer’s solution (Macopharma, France) and stored in ASC culture medium.

### 4.3. Morphometric Analysis of Alginate Microparticles

For morphometric analysis, the microparticles used contained ASCs from two donors at passage 3 (P3), and from one donor at passages 3 and 4 (P4). The alginate microparticles for each donor were examined by optical microscopy (ECLIPSE 50i, Nikon^®^) to assess their morphology. The diameters of 20 ASC-containing microparticles from the three donors (with one at P3 and P4) were measured on microscopy images using Image J software.

### 4.4. Viability and Metabolic Activity of Encapsulated Cells

To assess cell viability, ASC-containing alginate microparticles from two donors at P3, and one donor at P3 and P4 were dissolved by diluting one volume of alginate microparticles in two volumes of 35 mM sodium citrate solution (Sigma Aldrich, St. Louis, MO, USA) at pH = 7. This process did not interfere with cell viability. After a few seconds, the microparticles had totally dissolved, and the citrate was neutralized by adding PBS (phosphate buffered saline). The ASC were then centrifuged for 5 min at 1000 rpm. The cell pellet obtained was diluted 1/2 in 0.4% Trypan blue (Eurobio, France) and counted on a hemocytometer under optical microscopy (ECLIPSE 50i, Nikon^®^). Viability was assessed over the 16 days of maintenance of cells as microparticles. Every day, experiments were performed on three samples from each of the three donors.

Metabolic activity was measured using the MTT test. In this test, mitochondrial succinate dehydrogenase hydrolyses MTT (tetrazolium salt), forming crystals of formazan blue; the intensity of staining is an indication of cell viability. The MTT test was performed on encapsulated ASCs from two donors at P3, and from one donor at both P3 and P4. Microparticles were dissolved on D0, D2, D5, and D7 post-encapsulation and cells returned to monolayer culture. The metabolic activity of ASCs in monolayer cultures was also measured for comparison.

Cells were seeded on 6-well plates at 0.3 × 10^6^ cells/well. Each well containing ASCs was washed with lactated Ringer’s solution before adding 200 µL of MTT solution (Sigma Aldrich, Saint Quentin Fallavier, France) at 1 mg/mL. Plates were incubated for 2 h at 37 °C. MTT was then replaced by DMSO (WAK chemie medical GmbH, Germany) to dissolve the intracellular formazan blue crystals. Plates were incubated for 30 min under agitation. Optical density was measured at 450 and 550 nm. Experiments were performed on three samples from each of the three donors. ASC viability in DMEM was taken as the 100% viability control. A well treated with 0.05% SDS (Sigma Aldrich, St. Louis, MO, USA) was used as the cytotoxic positive control. For the assay to be valid, this well had to display less than 10% MTT activity. Results are reported as relative MTT activity compared to ASCs in DMEM.

### 4.5. Clonogenic Potential of Encapsulated ASCs

CFE (colony forming efficiency) is the reference parameter quantifying stem cells in solution. Clonogenic potential was assessed as follow: Cells were seeded in triplicate at low density in 6-well plates (200 cells/cm²) in proliferation medium, as above. Colonies were grown for 10 to 14 days, depending on the growth rate of the cells. At the end of the assay, culture plates were rinsed twice with PBS. Cells were then fixed with absolute ethanol for 5 min before staining with 20% GIEMSA (Sigma Aldrich, Saint Quentin Fallavier, France). Cell colonies were counted under optical microscopy. All three wells were counted for each donor. The final colony forming efficiency (CFE) value was determined by dividing the number of colonies counted ×100 by the number of cells seeded. Results are expressed as mean ± standard deviation for the replicates.

The clonogenic potential of encapsulated ASCs was tested on day 7 after encapsulation and compared to cells grown in monolayers for the three donors (two donors at P3 and one donor at P3 and P4).

### 4.6. Proliferative Potential of ASCs

Encapsulated cells and cells in monolayer were plated at 100,000 cells/well in 6-well plates and grown at 37 °C under humidified 5% carbon dioxide in the growth medium described above. Medium was replaced 1 h after plating to remove non-adherent cells [74] and subsequently every 2-3 days. Cells were detached at sub-confluency using trypsin-EDTA (Gibco, Thermo Fisher Scientific) and counted. The number of population doublings (PD), corresponding to the number of cell divisions that had occurred during the culture period, was calculated as follows: PD = log (N/ N0)/log 2, where N0 is the number of cells seeded, and N is the number of cells harvested. The doubling time (DT)—the number of cell divisions occurring each day—was calculated by dividing the PD by the culture duration in days.

The proliferative potential of ASCs was assessed using cells from two donors at P3, and one donor at both P3 and P4. Experiments were performed on three samples from each of the three donors.

### 4.7. Pluripotential

To assess the capacity of encapsulated ASC to differentiate into the three recommended lineages, ASCs were grown in differentiation media specific for each lineage. The pluripotent nature of cells was tested on encapsulated cells, cells from dissolved microparticles, and monolayer cultures as controls. After staining, cells were observed under optical transmitted light microscopy (ECLIPSE 50i, Nikon^®^). Experiments were performed in duplicate for each donor (two 6 well plates per donor).

For adipogenic differentiation, encapsulated ASCs from two donors at P3 were cultured in four wells from a 6-well plate for each donor. The remaining two wells on each plate were seeded with cells from the same donor from monolayer cultures, for comparison. Two of the four wells containing encapsulated cells and one of the wells containing monolayer cultures were filled with differentiation medium, and the other wells were filled with the culture medium described above. The adipocyte differentiation medium consisted of DMEM supplemented with 10% FCS, 10 µg/mL 3-Isobutyl-1-methylxanthine (IBMX), 100 µM indomethacin, 0.1 µM dexamethasone (all from Sigma Aldrich, Saint Quentin Fallavier, France), and 200 mUI insulin (Umulin, Lilly laboratories, Neuilly-sur-Seine, France). After 14 days, lipid droplets were stained with 0.4% Oil Red O after fixing cells in 10% formalin.

For osteogenic differentiation, ASCs from two donors at P3 were seeded in 6-well plates, as for adipogenic differentiation. Sub-confluent cells were induced using the StemPro^®^ Osteogenesis Differentiation Kit (Gibco, Life technologies, St Aubin, France). After 3 weeks, cells were fixed and stained with 40 mM Alizarin red (Merck Millipore, Fontenay sous Bois, France) to reveal calcium deposits.

Chondrogenic differentiation was assessed using the high-density pellet culture approach [75]. Sub-confluent cells from two donors at P3 were detached using trypsin-EDTA (Gibco, Thermo Fisher Scientific), counted, and 3.5 × 10^5^ viable ASCs were centrifuged (300 g, 10 min, twice) in a V-bottom 96-well plate to form a pellet. Pellets were treated for 21 days with a StemPro^®^ Chondrogenic Differentiation Kit (Gibco, Life technologies, St Aubin, France). On day 21, the pellet was harvested and fixed in 10% buffered formalin before embedding in paraffin. For histological analysis, tissue sections (5 µm) were deparaffinized, rehydrated, and chondrogenic differentiation was assessed by Alcian Blue staining of sulfated proteoglycans.

### 4.8. Characterization of Adipose-Derived Stem Cells

To verify that the encapsulation process did not alter the phenotype of encapsulated cells, the immunophenotypes of cells from donors 16,023 and 17,149 at P3 were assessed by flow cytometry the day after encapsulation and compared to those of cells from the same donors grown as monolayer. For these assays, microparticles were first dissolved in 35 mM sodium citrate, and released cells were isolated by centrifugation. Cells were resuspended in PBS at 0.5–1 × 10^6^ cells/mL. This cell suspension (100 μL) was stained using FITC-coupled CD45 and CD90 and PE-coupled CD73 and CD34 antibodies, or appropriate isotypic controls (All from BD Pharmigen, Le Pont de Claix, France). Antibodies were diluted in PBS. At least 10,000 events were acquired on a FACSCanto II cytometer (BD Biosciences, Le Pont de Claix, France) for analysis (DIVA 8.0.3 software).

### 4.9. Comparison of the Secretomes of Encapsulated and Monolayer ASCs

The secretome of encapsulated ASCs after 5 days of culture was compared to the secretome of ASCs grown as a confluent monolayer. Encapsulated and monolayer ASCs were incubated in serum-free-medium, supernatant was sampled at 48 h, and frozen at −80 °C. IL-6, IL-8, and VEGF were assayed by quantitative ELISA (R&D system, Minneapolis, MN, USA) in supernatants from cell cultures for seven donors at P3 with one donor at P3 and P4. A qualitative ELISA (Multi-Analyte ELISArray Kits, QIAGEN, Hilden, Germany) detecting 12 cytokines: IL-1β, IL-4, IL-6, IL-10, IL-12, IL-17A, IFNγ, TNFα, TGF-β1, MCP1, MIP 1α, and MIP 1β, was also performed on supernatants from four donors at P3. Experiments were performed on three samples from each donor.

### 4.10. Statistical Analyses

All experiments on viability, metabolic activity, clonogenic and proliferative potential, and cytokine detection by ELISA were performed in triplicate. Pluripotential assay was performed in duplicate and the microparticle size was calculated on 20 particles from each donor. Quantitative data obtained by measuring the microparticle size were assessed using the paired Wilcoxon test. For viability rates, MTT assays and quantitative data obtained by ELISA, colony forming efficiency, population doubling, and doubling time, statistical significance was calculated using the Mann and Whitney test. Statistical analyses were performed using GraphPad Prism 4.0 software (GraphPad Software Inc. V5.0c, La Jolla, CA, USA). All the data are expressed as means ± SEM. Statistically significant differences in figures are indicated by asterisks as follows: * *p* < 0.05, ** *p* < 0.001.

## Figures and Tables

**Figure 1 ijms-21-06316-f001:**
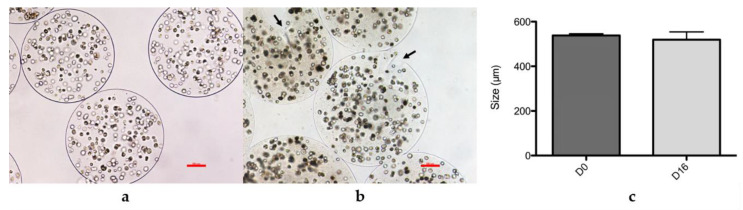
Morphology and size of microparticles at D0 and D16 after encapsulation. (**a**) Microparticles obtained for donor 16,023 at P3 at 4 × 10^6^ cells/mL encapsulation solution, D0, optical microscopy images, 10× magnification. Scale bar: 100 µm. (**b**) Microparticles obtained for donor 16,023 at P3 at 4 × 10^6^ cells/mL encapsulation solution, D16, optical microscopy images, 10× magnification. Scale bar: 100 µm. (**c**) Mean diameters ± standard deviation of microparticles from two donors at P3 and one donor at P3 and P4 (*n* = 4) measured after encapsulation (D0) and after 16 days’ (D16) storage in culture medium. Diameters were calibrated using a micrometric ruler and were measured on photos taken with the optical microscope, using Image J software.

**Figure 2 ijms-21-06316-f002:**
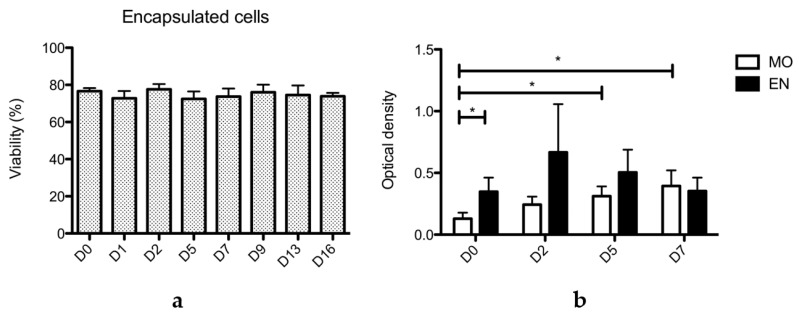
Viability and proliferation of encapsulated and monolayer ASCs from the three donors (16,023 at P3 and P4, 16,198 and 16,148 at P3, *n* = 4). (**a**) Viability of encapsulated ASCs numerated after Trypan blue coloration from the day of encapsulation (D0) until 16 days after encapsulation (D16) on 500-µL aliquots. Viability of encapsulated cells remains unchanged for 16 days. (**b**) MTT assays performed on ASCs from the three donors grown as monolayers (MO) and encapsulated (EN). Assays were performed on D0, D2, D5, and D7 after encapsulation. Comparison of the optical density shows a significative difference between D0 and D5 and D0 and D7 for MO cells, a sign of both proliferation and an increase of metabolic activity. Comparison of the two conditions (MO and EN) shows a significative difference only at D0. Statistical significance was assessed with the Mann and Whitney test. *: *p* < 0.05.

**Figure 3 ijms-21-06316-f003:**
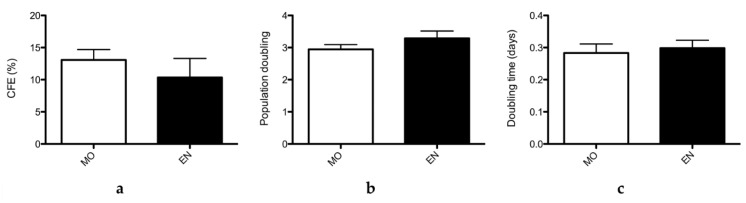
Stem cell properties of encapsulated cells (donors 16,023 at P3 and P4, 16,198 and at P3, *n* = 4). (**a**) CFE (colony forming efficiency) determined for the three donors in the MO (monolayer) condition and in the EN condition (encapsulated for 7 days then re-seeded as a monolayer after dissolving the microparticles). (**b**) Population doubling for ASCs from the three donors in the monolayer (MO) or encapsulated (EN) conditions. (**c**) Doubling time for ASCs from the three donors in the monolayer (MO) or encapsulated (EN) conditions.

**Figure 4 ijms-21-06316-f004:**
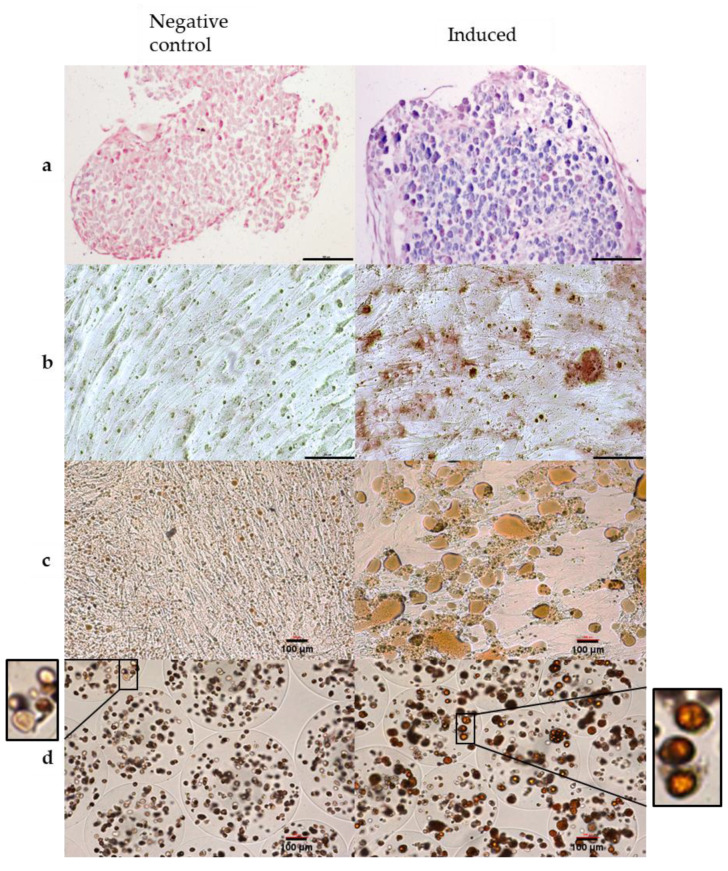
In vitro differentiation capacity of encapsulated ASCs assessed for the three recommended mesenchyme lineages. Scale bar: 100 µm, 10× magnification, optical microscopy with transmitted light. (**a**) Cells in condition 2, untreated (control) or treated (induced) with a chondrogenic induction medium for 21 days and stained with Alcian Blue which color in blue acid mucosubstances and acetic mucins. (**b**) cells in condition 2, untreated (control) or treated (induced) with an osteogenic induction medium for 21 days and stained with Alizarin Red, which color in red calcium deposit. (**c**) cells in condition 2, untreated (control) or treated (induced) with an adipogenic induction medium for 14 days and stained with Oil red O, which color lipids in red. (**d**) cells in condition 1, untreated (control) or treated (induced) with an adipogenic induction medium for 14 days and stained with Oil red O, which color lipids in red. The blow-up from the original picture on the sides show lipid vacuoles formed after ASC induction into adipocytes (on the right) and the absence of these vacuoles in the control sample (on the left).

**Figure 5 ijms-21-06316-f005:**
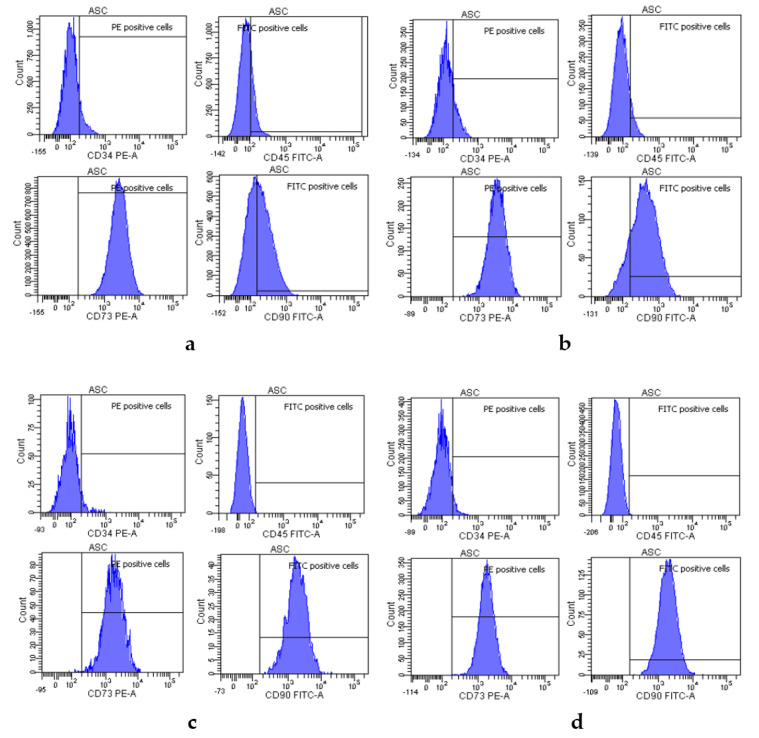
ASC characterization based on surface marker profiles, detected by flow cytometry. (**a**) Immunophenotyping of ASCs encapsulated from donor 16,023 at P3. (**b**) Immunophenotyping of ASCs encapsulated from donor 17,149 at P3. (**c**) Immunophenotyping of monolayer ASCs from donor 16,023 at P3. (**d**) Immunophenotyping of monolayer ASCs from donor 17,149 at P3.

**Figure 6 ijms-21-06316-f006:**
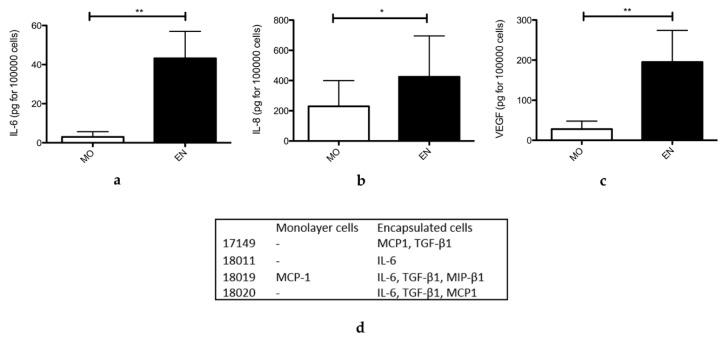
Secretome of encapsulated and monolayer ASC cultures detected by quantitative ELISA in supernatants sampled after 48 h culture in serum-free medium for donors 16,023 at P3 and P4, 16,148, 18,011, 18,019, 18,020, 16,198, and 17,149 at P3, *n* = 8 (**a**–**c**). Cells were grown as monolayers (MO) or were encapsulated (EN). Statistical significance of unpaired comparisons was assessed with the Mann and Whitney test. *: *p* < 0.05; **: *p* < 0.01 (**a**) Concentrations of cytokine IL-6 expressed in pg for 100,000 cells obtained for monolayer cells and encapsulated cells. (**b**) Concentrations of cytokine IL-8 expressed in pg for 100,000 cells obtained for monolayer cells and encapsulated cells. (**c**) Concentrations of VEGF expressed in pg for 100,000 cells obtained for monolayer cells and encapsulated cells. (**d**) Cytokines detected by qualitative ELISA in supernatants sampled after 48 h culture in serum-free medium for four donors (17,149, 18,011, 18,019, and 18,020 at P3, *n* = 4). Cells were grown as monolayers or were encapsulated.

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
