# Peer review of "Encapsulation of Adipose-Derived Mesenchymal Stem Cells in Calcium Alginate Maintains Clonogenicity and Enhances their Secretory Profile"

_ijms, 2020, doi:10.3390/ijms21176316_

Round 1
Reviewer 1 Report
The authors examined the effect of alginate encapsulation of adipose-derived MSCs (ASCs) in vitro. They conclude that encapsulation does not negatively affect the cells and increases their production of certain factors. There are some comments to be addressed concerning the sample size calculations and the need for additional comparisons between monolayer and encapsulated cells.
- A major concern is the N for each experiment. For example, when assessing the diameter of the microparticles (Lines 88-92 and Figure 1c) 20 microparticles from each of 4 samples were measured. The N for this experiment is reported as 80, but should be reported as 4, each with 20 replicates. Statistics should be re-calculated accordingly.
- Line 115-116: The authors state that the metabolic activity measured for monolayer cells increased regularly from D0 to D7, indicating that cells were proliferating. However, while D0 is significantly different from D5 & D7, Days 2, 5 and 7 are not indicated as being different from each other. In Fig 2c, there is no indication that the optical density is different at any of the time points. Please graph panels b & c on 1 graph and assess if the optical density of the 2 conditions is different.
- In Fig 3 the authors state, “n=3 for each donor” does that mean the experiment was done 3 times with P3 or P4 cells, as appropriate, or that the experiment was done in triplicate and each replicate represents n=1, the latter of which is not appropriate.
- For Figure 4, the authors examined encapsulated cells in duplicate but monolayer cells in single wells – all studies should be done in at least duplicate. Were there differences in differentiation between monolayer and encapsulated cells. In Fig 4d is the inset a blow up of the 10x micrograph or a new micrograph using a higher magnification?
- In Figure 5, please compare the cell surface profile of monolayer and encapsulated cells.
- What is the n for Fig. 6? Is it replicates or individual wells?
- In line 266-268, the authors state, “our results showed that encapsulated ASC differentiated into mainly unilocular adipocytes, like those found in human adipose tissue [49], in contrast to ASC grown as a monolayer, which produce multilocular adipocytes”; however, they do not demonstrate staining of monolayer cells or quantify differences.
Author Response
Dear reviewer 1,
Please find herewith the following manuscript, “Encapsulation of adipose derived mesenchymal stem cells in calcium alginate maintains clonogenicity and enhances their secretory profile”, by “Lucille Capin, Nacira Abbassi, Maelle Lachat, Marie Calteau, Cynthia Barratier, Ali Mojallal, Sandrine Bourgeois, Céline Auxenfans”, which we wish to submit for publication as an original article in “International Journal of Molecular Sciences”. I would like to thank you for the comments you have made and the time you took to review this work. We took into account all of your comments and according to them we have modified in the article.
Below is a point-by-point answer to the issues you raised.
- A major concern is the N for each experiment. For example, when assessing the diameter of the microparticles (Lines 88-92 and Figure 1c) 20 microparticles from each of 4 samples were measured. The N for this experiment is reported as 80, but should be reported as 4, each with 20 replicates. Statistics should be re-calculated accordingly.
The N concerning figure 1c has been changed by 4 and the statistics were re-calculated accordingly with the paired Wilcoxon test (lines 89-93). The caption of the figure and the figure 1c itself have been modified too (lines 101-102).
Figure 1. Morphology and size of microparticles at D0 and D16 after encapsulation. (a) Microparticles obtained for donor 16023 at P3 at 4x106 cells/mL encapsulation solution, D0, optical microscopy images, 10x magnification. Scale bar: 100 µm. (b) Microparticles obtained for donor 16023 at P3 at 4x106 cells/mL encapsulation solution, D16, optical microscopy images, 10x magnification. Scale bar: 100 µm. (c) Mean diameters ± standard deviation of microparticles from two donors at P3 and one donor at P3 and P4 (n=4) measured after encapsulation (D0) and after 16 days’ (D16) storage in culture medium. Diameters were calibrated using a micrometric ruler and were measured on photos taken with the optical microscope, using Image J software.
We have therefore specified all the “n” in all figures caption and we have clarified the number of replicate in material and method section for all the experiments and in statistical analysis section.
- Line 115-116: The authors state that the metabolic activity measured for monolayer cells increased regularly from D0 to D7, indicating that cells were proliferating. However, while D0 is significantly different from D5 & D7, Days 2, 5 and 7 are not indicated as being different from each other. In Fig 2c, there is no indication that the optical density is different at any of the time points. Please graph panels b & c on 1 graph and assess if the optical density of the 2 conditions is different.
We have grouped figure 2 b and c to have one figure showing MTT test for both monolayer cells and encapsulated cells (you can see the revised figure 2 below). Indeed, we have observed a statistical difference for monolayer cells between D0 and D5 et D0 and D7 but not between D2 and D5, D2 and D7 and D5 and D7 so we have changed our statement line 116-117 and remove “regularly”. Statistical analysis of optical density for encapsulated cells have shown no difference between all the time point (D0 and D2, 5, 7, D2 and D5, 7 and D5 and D7). As you have suggested, we also compared the optical density between the two condition (MO vs EN) and we have only found one statistical difference at D0. We therefore modified the text above (lines 119-120, 124-127) and the figure and caption (line 130, 133, 135-140).
Figure 2. Viability and proliferation of encapsulated and monolayer ASC from the three donors (16023 at P3 and P4, 16198 and 16148 at P3, n=4). (a) Viability of encapsulated ASC numerated after Trypan blue coloration from the day of encapsulation (D0) until 16 days after encapsulation (D16) on 500 µL aliquots. Viability of encapsulated cells remains unchanged for 16 days. (b) MTT assays performed on ASC from the three donors grown as monolayers (MO) and encapsulated (EN). Assays were performed on D0, D2, D5, and D7 after encapsulation. Comparison of the optical density shows a significative difference between D0 and D5 and D0 and D7 for MO cells sign of both proliferation and increase of metabolic activity. Comparison of the two conditions (MO and EN) shows a significative difference only at D0. Statistical significance was assessed with the Mann and Whitney test. *: p < 0.05.
- In Fig 3 the authors state, “n=3 for each donor” does that mean the experiment was done 3 times with P3 or P4 cells, as appropriate, or that the experiment was done in triplicate and each replicate represents n=1, the latter of which is not appropriate.
Experiments in figure 3 have been done on three donors at P3 with one donor at P3 and P4 (n=4). All experiments were performed in triplicate for each donor. We therefore remove “n=3” in figure 3 caption (line 170) which was not appropriate and replace it line 166 by n=4.
- For Figure 4, the authors examined encapsulated cells in duplicate but monolayer cells in single wells – all studies should be done in at least duplicate. Were there differences in differentiation between monolayer and encapsulated cells. In Fig 4d is the inset a blow up of the 10x micrograph or a new micrograph using a higher magnification?
Experiment on pluripotential was performed in duplicate for each donor. We used two 6 wells plates for each donor with, for one 6 wells plate : two wells for encapsulated cells induced, two wells for encapsulated cells control, one well for monolayer cells induced and one well for monolayer cells control. We clarified it line 469 in material and method section.
In figure 4d the inset is a blow up of the 10x picture. We have specified it in the figure 4 caption line 203.
- In Figure 5, please compare the cell surface profile of monolayer and encapsulated cells.
As you asked, we performed a flow cytometry on monolayer cells from the same donors and we have compared cell surface profile of the two conditions in figure 5 (below). Figure 5, the text above (lines 208-209 and 212-213) and caption (lines 217-219) have been modified.
Figure 5. ASC characterization based on surface marker profiles, detected by flow cytometry. (a) Immunophenotyping of ASC encapsulated from donor 16023 at P3. (b) Immunophenotyping of ASC encapsulated from donor 17149 at P3. (c) Immunophenotyping of monolayer ASC from donor 16023 at P3. (d) Immunophenotyping of monolayer ASC from donor 17149 at P3.
- What is the n for Fig. 6? Is it replicates or individual wells?
For figure 6 a,b,c secretomes from six donors at P3 and one donor at P3 and P4 were assessed in triplicate by quantitative ELISA, so n = 8. We have specified it in the caption line 247. For figure 6d, secretome from four donors at P3 were analysed by qualitative ELISA, so n = 4. We have specified it in caption line 254.
- In line 266-268, the authors state, “our results showed that encapsulated ASC differentiated into mainly unilocular adipocytes, like those found in human adipose tissue [49], in contrast to ASC grown as a monolayer, which produce multilocular adipocytes”; however, they do not demonstrate staining of monolayer cells or quantify differences
We don’t have enough pictures of adipocyte differentiation to quantify differences. We only see on photos from figure 4c and d that the lipid vacuoles seems to be unilocular on encapsulated cells (figure 4d) but multilocular on monolayer cells (figure 4c). As you have pointed out, without quantification we can’t affirm this data so we have removed this sentence from results lines 188-192 and from discussion lines 277-279.
We are looking forward to hearing from you, and we hope modifications that we have made will suit you and will allow our manuscript for publishing.
Yours sincerely,
Lucille Capin

Reviewer 2 Report
Rev 1 has addressed most of previous concerns.
However, some additional effort should be done on Figure 4, 5 and 6 in order to make results more readable.
Figure 4: in the result section you describe two experimental conditions (condition 1 and 2). However, in the images provided in figure 4 there is no clear association with the two experimental conditions. Please clarify.
Figure 5: please specify the experimental condition; has flow cytometry been perfomed after that the encapsulated cells were removed from algynate ?
Figure 6: in figure 6b you describe the effect of encapsulation on IL-8 release in the supernatant; however, figure 6d does not report this cytokine among those detected by qualitative ELISA in supernatants sampled after 48 h culture. Please clarify.
Author Response
Dear reviewer 2,
Please find herewith the following manuscript, “Encapsulation of adipose derived mesenchymal stem cells in calcium alginate maintains clonogenicity and enhances their secretory profile”, by “Lucille Capin, Nacira Abbassi, Maelle Lachat, Marie Calteau, Cynthia Barratier, Ali Mojallal, Sandrine Bourgeois, Céline Auxenfans”, which we wish to submit for publication as an original article in “International Journal of Molecular Sciences”. I would like to thank you for the comments you have made and the time you took to review this work. We took into account all of your comments and according to them we have modified in the article.
Below is a point-by-point answer to the issues you raised.
- Figure 4: in the result section you describe two experimental conditions (condition 1 and 2). However, in the images provided in figure 4 there is no clear association with the two experimental conditions. Please clarify.
To be more clear, we have added the different experimental condition in figure 4 caption (lines 196-201) :
Figure 4. In vitro differentiation capacity of encapsulated ASC assessed for the three recommended mesenchyme lineages. Scale bar: 100 µm, 10x magnification, optical microscopy with transmitted light. (a) Cells in condition 2, untreated (control) or treated (induced) with a chondrogenic induction medium for 21 days and stained with Alcian Blue which color in blue acid mucosubstances and acetic mucins. (b) cells in condition 2, untreated (control) or treated (induced) with an osteogenic induction medium for 21 days and stained with Alizarin Red which color in red calcium deposit. (c) cells in condition 2, untreated (control) or treated (induced) with an adipogenic induction medium for 14 days and stained with Oil red O which color lipids in red. (d) cells in condition 1, untreated (control) or treated (induced) with an adipogenic induction medium for 14 days and stained with Oil red O which color lipids in red. The blow-up from the original picture on the sides show lipid vacuoles formed after ASC induction into adipocytes (on the right) and absence of these vacuoles in control sample (on the left).
- Figure 5: please specify the experimental condition; has flow cytometry been perfomed after that the encapsulated cells were removed from algynate ?
Yes, flow cytometry has been perfomed after that the encapsulated cells were removed from alginate. We have specified it in material and method section (lines 500-501) and also in the text above figure 5 lines 208-209.
- Figure 6: in figure 6b you describe the effect of encapsulation on IL-8 release in the supernatant; however, figure 6d does not report this cytokine among those detected by qualitative ELISA in supernatants sampled after 48 h culture. Please clarify
Indeed, the multiplex qualitative ELISA is able to detect 12 cytokines but IL-8 is not one of them. We specified it in the text above figure 6 lines 234-235 and we have detailed cytokines that this test was able to detect in materiel and method section lines 511-513.
We are looking forward to hearing from you, and we hope modifications that we have made will suit you and will allow our manuscript for publishing.
Yours sincerely,
Lucille Capin